# Genome-Wide Analysis and Identification of the *Aux/IAA* Gene Family in Peach

**DOI:** 10.3390/ijms20194703

**Published:** 2019-09-23

**Authors:** Dan Guan, Xiao Hu, Donghui Diao, Fang Wang, Yueping Liu

**Affiliations:** 1Beijing Key Laboratory of New Technique in Agricultural Application, Beijing University of Agriculture, Beijing 102206, China; guanadan@hotmail.com (D.G.); hxiao_0323@sina.com (X.H.); 2College of Plant Science and Technology, Beijing University of Agriculture, Beijing 102206, China; 3College of Bioscience and Resources Environment, Beijing University of Agriculture, Beijing 102206, China; 18210853628@163.com; 4Food science and Engineering College, Beijing University of Agriculture, Beijing 102206, China; 5Key Laboratory for Northern Urban Agriculture Ministry of Agriculture and Rural Affairs, Beijing University of Agriculture, Beijing 102206, China

**Keywords:** peach, auxin, *Aux/IAA* genes, expression analysis

## Abstract

The Auxin/indole-3-acetic acid (*Aux/IAA*) repressor genes down-regulate the auxin response pathway during many stages of plant and fruit development. In order to determine if and how *Aux/IAAs* participate in governing texture and hardness in stone fruit maturation, we identified 23 *Aux/IAA* genes in peach, confirmed by the presence of four conserved domains. In this work, we used fluorescence microscopy with PpIAA-GFP fusion reporters to observe their nuclear localization. We then conducted PCR-based differential expression analysis in “melting” and “stony hard” varieties of peach, and found that in the “melting” variety, nine *PpIAAs* exhibited peak expression in the S4-3 stage of fruit maturation, with *PpIAA33* showing the highest (>120-fold) induction. The expression of six *PpIAAs* peaked in the S4-2 stage, with *PpIAA14* expressed the most highly. Only *PpIAA15/16* showed higher expression in the “stony hard” variety than in the “melting” variety, both peaking in the S3 stage. In contrast, *PpIAA32* had the highest relative expression in buds, flowers, young and mature leaves, and roots. Our study provides insights into the expression patterns of *Aux/IAA* developmental regulators in response to auxin during fruit maturation, thus providing insight into their potential development as useful markers for quantitative traits associated with fruit phenotype.

## 1. Introduction

Peach (*Prunuspersica* L. Batsch) is a typical climacteric crop that has been bred for a variety of fruit traits, especially firmness. Mature peaches can be divided into three phenotypes based on the softening of the mesocarp: “melting”, “non-melting”, and “stony hard”. The melting phenotype is characterized by increasing softness with maturity caused by the activity of endopolygalacturonase during the advanced stages of ripening [1]. Non-melting peaches soften slowly when overripe and never melt, whereas fruits of the stony hard type exhibit very firm and crisp flesh at the ripened stage [2,3]. Previous research has shown that regulation of ethylene biosynthesis depends on high local concentrations of indole-3-acetic acid (IAA) in the peach mesocarp, and thus peaches with the non-softening or hard phenotype exhibit significantly lower levels of both IAA and downstream ethylene synthesis than melting type fruits [3].

IAA is a member of the auxin family of phytohormones, and is ubiquitously involved in various biological processes, such as organ development, phototropism, response to environmental stimuli, and fruit ripening [4]. The perception and transduction of auxin signals involves the cooperative action of several components, among which, auxin/indole-3-acetic acid (Aux/IAA) proteins play a pivotal role. The *Aux/IAA* genes represent a classical auxin-responsive gene family whose members typically undergo rapid induction by auxin. Aux/IAAs are nuclear proteins with short half-lives that generally contain four conserved domains [5]. Domain I contains the leucine repeat motif ‘LxLxLx’ that represses transcription via interaction with the TOPLESS (TPL) co-repressor [6,7]. Domain II is conserved, plays a major role in protein stability, and is required for auxin-regulated signaling via interaction with a component of the transport inhibitor response 1 (TIR1) degradation pathway [8]. Domains III and IV, which are also found in auxin response factor (ARF) proteins, are required for homodimerization among Aux/IAA family members and heterodimerization between Aux/IAA and ARF proteins [7,9,10]. Moreover, Aux/IAAs have two nuclear localization signals (NLSs) that are responsible for protein targeting to the nucleus [11,12].

Previous research on the role of *Aux/IAA* family genes has shown that they make essential regulatory contributions to many aspects of plant development, such as adventitious root initiation, apical dominance, and fruit development [13,14,15]. Several *Aux/IAA* genes across multiple species have been shown to contribute to fruit development and maturation. In strawberry (*Fragaria x ananassa*), the expression of two auxin-responsive genes, *FaAux/IAA1* and *FaAux/IAA2*, was highly up-regulated during early fruit development, but declined steeply during ripening [16]. In tomato (*Solanum lycopersicum*), *SlIAA3*, an ethylene- and auxin-responsive gene, was found to be expressed in all tissues, but was most highly expressed in orange, red, and ripe fruits in the late stages of fruit ripening [17].

Other *Aux/IAA* genes were found to participate in fruit maturation. For example, in tomato, mutation of *SlIAA9* led to morphological changes in plant development, including simple, rather than compound, leaves, increased height and internode development, longer roots, and importantly, prefertilization fruit set and parthenocarpy [16,18,19]. In a study by Su, RNAi silencing of the *Aux/IAA* repressor *SlIAA17* led to enlarged tomato fruit compared to wild type, caused by dysregulation of endoreduplication that resulted in hyper-expanded pericarp cells [20]. Similarly, expression of peach *PpIAA19* in tomato led to increased plant height and number of lateral roots, as well as parthenocarpy and altered fruit morphology [21]. Clarifying the genetic and hormonal mechanisms underlying control of fruit maturation has thus been a long-standing challenge among geneticists, plant breeders, and fruit producers.

While the majority of previous fruit-related research on *Aux/IAAs* has focused on early stages of fruit growth and development [19,22,23], relatively little research has explored the potential roles of *Aux/IAA* expression in determining fruit phenotypes during the ripening stage. In this study, to obtain the possible genes for auxin regulation of peach fruit ripening, two types of ripening peach fruits, a “melting” variety, ‘Okubo’, and a “stony-hard” variety, ‘Jing Yu’ were used to perform a comparative analysis of *PpIAA* genes between the two phenotypes, which included identification of the members of the entire *PpIAA* gene family, analysis of their gene structures, and evaluation of the phylogenetic relationships between the *PpIAAs*, *AtIAAs*, and *SlIAAs*. Tandem Mass spectrometry and gas chromatography were used to quantify IAA and ethylene production at different stages of fruit development and ripening. Quantitative real-time PCR (qRT-PCR) was used to analyze the expression patterns of *PpIAA* genes during fruit development, in various tissues, and in response to treatment with the auxins NAA (1-naphthylacetic acid) and PCIB (p-chlorophenoxyisobutyric acid). The goal of this work was to determine if and how *PpIAAs* were differentially expressed during fruit ripening, thus implicating them as candidates for future functional analysis or as targets for potential development in molecular breeding. Our results establish a foundational basis for further functional molecular and biochemical characterization of *Aux/IAA* activity in peach, thus extending the current understanding of auxin signaling during fruit development. Our advances in auxin signaling will not only prove useful for understanding the basic genetics and reproductive physiology of tree fruits, but will also provide tools for engineering and breeding for novel traits in peach and other stone fruits.

## 2. Results

### 2.1. PpIAAs Identified in Peach Genome

To identify members of the *Aux/IAA* gene family in peach, we performed BLAST searches using the entire amino acid sequences of all four conserved Aux/IAA domains of the *Arabidopsis* protein as a query sequence against the GDR database. A total of 23 Aux/IAA sequences were found in the peach genome. Information about these 23 *PpIAA* genes is listed in Table 1, including the gene name, Genbank ID, peach gene ID, location, open reading frame (ORF) length, protein length, molecular weight (MW), and isoelectric point (pI). The size of the predicted PpIAAs ranged from 162 amino acids (PpIAA33) to 413 amino acids (PpIAA12), with MWs ranging from 17.98 kDa (PpIAA33) to 45.66 kDa (PpIAA8). The predicted pIs varied widely, from 4.89 (PpIAA32) to 9.30 (PpIAA29).

### 2.2. Proteins Are Separated into Four Phylogenetic Clades

A multiple sequence alignment was constructed using amino acid sequences of the 23 PpIAAs. Four conserved domains were identified (I, II, III, and IV) (Figure 1). We found that 21 of the PpIAA family members share all four conserved domains, while PpIAA26 lacks domains I and II, and PpIAA33 lacks domain I. Most PpIAAs were found to contain nuclear localization signals (NLSs). The typical NLS, also called an SV40-type NLS, is located at the end of domain IV. The βαα motif (one β sheet and two α helices), which functions in the dimerization of Aux/IAAs, was also found within domain III in a majority of the PpIAAs.

To examine the evolutionary relationships among the Aux/IAAs from *Arabidopsis*, tomato (*Solanum lycopersicum* L.), and peach (*Prunus persica* L. Batch), a rooted phylogenetic tree was generated based on the alignment of amino acid sequences of 82 Aux/IAAs, including 34 from *Arabidopsis*, 25 from tomato, and 23 from peach. The phylogenetic tree could be divided into group I, group II, and group III, and group I could further be divided into four subgroups (I-a, I-b, I-c, I-d) (Figure 2). In this phylogenetic tree, a total of 17 sister pairs were found, including 6 AtIAA-AtIAA pairs, 5 SlIAA-SlIAA Pairs, 1 PpIAA-PpIAA pairs, 3 SlIAA-PpIAA pairs, and 2 AtIAA-PpIAA pairs. The peach family (23 members) is slightly contracted compared with the size of that of *Arabidopsis* (34 members) and tomato (25 members). With reference to *Arabidopsis,* four clades (I-a, I-d, II and III) are contracted in the peach and one (I-b) is expanded. With reference to tomato, two clades (I-a and I-c) are contracted in the peach and two (I-d and III) are expanded. A sister pair (PpIAA13 and PpIAA29) was identified in group III. Our results support previous observations regarding *Aux/IAA* gene duplication [24].

### 2.3. Representative PpIAAs are Localized to the Nucleus

To confirm the subcellular localization of PpIAAs, online protein prediction software (http://www.predictprotein.org/home) was used to predict the subcellular localization of PpIAAs [25]. The results showed that all PpIAAs were predicted to localize to the nucleus (Figure 1). Six PpIAA proteins, PpIAA1, PpIAA5, PpIAA9, PpIAA11, PpIAA13 and PpIAA14, which represent the different peach *Aux/IAA* subclades, were chosen for further analysis of their subcellular localization. For expression of GFP-tagged proteins, 35S::PpIAA-EGFP dual-expression vectors were simultaneously constructed for each PpIAA, and a 35S::EGFP construct was used as a positive control. The EGFP signals of the PpIAA1, PpIAA5, PpIAA9, PpIAA11, PpIAA13 and PpIAA14 fusion proteins were only observed in the nucleus, with a strong green fluorescent signal (Figure 3). These fluorescent microscopy data thus confirm that the selected PpIAAs are nucleoproteins based on their targeting and translocation to the nucleus.

### 2.4. Significantly Higher IAA Concentration and Ethylene Production in Melting Cultivar Compared to Stony Hard Cultivar Inversely Related to Flesh Firmness During Peach Fruit Development and Ripening

To monitor changes in IAA concentration, ethylene production, and fruit firmness during development and ripening within the two cultivars, ‘Okubo’ and ‘Jing Yu’ fruits were sampled from the S1 stage to the S4-3 stage (Figure 4). The IAA concentrations were highest in young fruits and then gradually decreased during development, reaching a minimum at the S2 stage. In ‘Okubo’ fruit, IAA levels sharply increased during the S4-1 stage and remained high in the following stages. In contrast, in ‘Jing Yu’ fruit, IAA levels were lower than in all developmental stages of ‘Okubo’ (Figure 4A). Ethylene production in ‘Okubo’ fruit sharply increased between the S4-1 and S4-3 stages, while this sharp increase was not observed in ‘Jing Yu’ fruit (Figure 4B). In both cultivars, flesh firmness gradually decreased during fruit development and ripening, although the firmness of ‘Jing Yu’ fruit was significantly greater than in ‘Okubo’ fruit during the S4-2 and S4-3 stages (Figure 4C).

### 2.5. PpIAAs are Differentially Expressed within Specific Tissues

In order to characterize the expression patterns of *PpIAA* genes across different tissues, and potentially identify expression patterns related to fruit ripening and firmness, the relative expression levels of all *PpIAAs* were analyzed in the ‘Okubo’ melting cultivar, including fruit in the S1 stage, fruit in the S4-3 stage, as well as in buds, flowers, young leaves, mature leaves, and roots (Figure 5). These assays revealed that some *PpIAAs* showed tissue-specific expression patterns. 

The expression of *PpIAA13*, *PpIAA14* and *PpIAA33* genes in S4-3 fruit were transcriptionally induced more than 10-fold higher than that in S1 fruit, especially *PpIAA33* (~130-fold increase), suggesting that *PpIAA33* plays a necessary role in fruit development and ripening. Only *PpIAA32* exhibited a >5-fold increase in expression levels in both buds and flowers. *PpIAA14* also increased >5-fold in buds, and *PpIAA18* and *PpIAA20* also showed a >5~10-fold increase in flowers. However, the majority of genes, including *PpIAA3/5/8/9/11/12/15/16/17/26/27/27′* were all substantially down-regulated (fold change <1) in both buds and flowers. *PpIAA32* was also found to be highly up-regulated in both young and mature leaves. Similar to observations in buds and flowers, transcription of the majority of *PpIAAs* was down-regulated in leaf tissue. In roots, however, expression of *PpIAA13/14/18/20/26/29/32* was >10 times higher than that in S1 fruit, and especially *PpIAA29* and *PpIAA32* exhibited a >100-fold induction (Figure 5).

### 2.6. Expression of PpIAA Genes in Fruits of Two Cultivars at Different Stages

In order to identify which specific *PpIAA* genes may have the highest expression, and thus potentially the strongest regulatory influence on fruit development, we analyzed the expression levels of *PpIAAs* in both the melting ‘Okubo’ and the stony hard ‘Jing Yu’ fruits at each developmental stage from cell division and enlargement (S1) through fruit maturity (S4-3) (Figure 6 and Figure 7). The 23 *PpIAA* genes were classified into groups A–D according to their changes of expression levels at different developmental stages in two varieties. Group A could be further divided into two subgroups according to the genes expression level at S4 (A-1 and A-2). 

The 14 genes in group A were more highly expressed in ‘Okubo’ than in ‘Jing Yu’ in almost all developmental stages. In group A-1, expression of those genes continuously increased in the ‘Okubo’ fruit from stage S4-1 to S4-3 and reached peak expression in the S4-3 stage. The mRNA levels of genes in the A-2 group reached peak expression in S4-1 or S4-2 and finally decreased in S4-3 in ‘Okubo’ fruit (Figure 6A). Given their high expression in the melting variety, it is likely that expression of these genes contributes to the softening of the mesocarp during ripening.

Group B is comprised of two genes more highly expressed in the stony hard cultivar during ripening stages. *PpIAA15* and *PpIAA16* exhibited higher expression levels in ‘Jing Yu’ than in ‘Okubo’ fruit from S3 to S4-3. *PpIAA15* expression showed 280- and 25-fold in ‘Jing Yu’ than in ‘Okubo’ at S3 and S4-3, but *PpIAA16* expression showed 3-fold in ‘Jing Yu’ than in ‘Okubo’ at S3 (Figure 6B). Given their higher expression in the stony hard phenotype, it is possible that expression of these genes contributes to enhanced firmness in mature fruits.

Group C was comprised of 6 genes that displayed lower expression level in all fruit developmental stages. In S2, the pit hardening stage, *PpIAA3*, *PpIAA12* and *PpIAA27* expression level was higher in ‘Jing Yu’ than in ‘Okubo’ (Figure 6C-1), but *PpIAA7*, *PpIAA18* and *PpIAA27’* expression levels were lower in ‘Jing Yu’ than in ‘Okubo’ (Figure 6C-2). These genes possibly related with the early development of fruit in peach. Additionally, in group D, the expression of *PpIAA8* was not significantly different between the two cultivars at any stage (Figure 6D).

Cluster analysis (Figure 7) showed that *PpIAA14* and *PpIAA33* were both highly up-regulated in all melting variety samples, though *PpIAA14* peaked at S4-2, while *PpIAA33* had its lowest significantly different expression at S4-2 and two peaks in expression at S2 and S4-3, while in ‘Jing Yu’ these genes are both down-regulated. In addition to these two genes, it is noteworthy that in the melting variety, ‘Okubo’, 10 of the assayed genes are up-regulated in the S4-3 stage, and the rest are comparable with the control (neither up- nor down-regulated), while in the stony hard ‘Jing Yu’, only *PpIAA15* is transcriptionally induced (though not in ‘Okubo’). In light of the dramatic phenotypic differences related to mesocarp texture, these data suggest that up-regulation of this suite of *PpIAAs* in ‘Okubo’ may strongly contribute to the softened fruit, while their down-regulation in ‘Jing Yu’ contributes to the hard fruit phenotype.

### 2.7. Expression Analysis of PpIAA Genes under 1-Naphthylacetic Acid (NAA) and P-Chlorophenoxyisobutyric Acid (PCIB) Treatments

As an essential, primary auxin-responsive gene family, the transcriptional regulatory response to auxin treatment by *Aux/IAA* genes is rapid. To verify that *PpIAA* gene expression is up-regulated following treatment with the auxin analogs 1-naphthylacetic acid (NAA) and p-chlorophenoxyisobutyric acid (PCIB), qRT-PCR was performed with total RNA isolated from the mesocarp of stony hard ‘Jing Yu’ fruit during the S4-1 stage. We found that expression of most *PpIAA* genes was induced by NAA when compared to their expression in untreated controls. In particular, *PpIAA1*, *PpIAA8* and *PpIAA17* showed strong sensitivity to NAA, with a >10-fold increase in expression across all time points, while *PpIAA5* and *PpIAA29* exhibited relatively modest up-regulation, if any, only crossing to >10-fold induction after 12 h of exposure to NAA (Figure 8). Notably, *PpIAA17* transcription was induced by >200-fold at 1.5 h after NAA treatment and maintained this level of expression for the duration of the time course. In contrast, treatment with PCIB led to minimal or no significant change in expression compared to untreated controls. These data show that *PpIAA1/8/17* respond quickly to auxin treatment (1.5 h) and indicating that these are auxin early response genes.

## 3. Discussion

*Aux/IAA* proteins are critical auxin-mediated developmental signal transduction, and in the absence of auxin, these proteins bind to ARF (auxin response factor) transcription factors to repress transcription of target genes [26,27]. High auxin levels activate theTIR1/AFB (TRANSPORT INHIBITOR RESPONSE 1/AUXIN SIGNALING F-BOX) receptors, subsequently leading to ubiquitination of Aux/IAA proteins and degradation via the 26S proteasome [18,28,29]. The ARFs are thus released to up-regulate the downstream auxin response genes [28].

*Aux/IAA* gene families have been identified and analyzed in many diverse plant species, including *A. thaliana* (34 members) [10], rice (31 members) [30], *B. napus* (119 numbers) [24], tomato (25 members) [12], and papaya (18 members) [31]. In this study, a comprehensive set of 23 *Aux/IAA* genes were identified and characterized from the GDR peach genome database. We found that between any two PpIAAs, amino acid sequence similarity typically falls between 50–70%, with the lowest similarity observed between PpIAA27 and PpIAA27′ (28.7%) and the highest similarity observed between PpIAA3 and PpIAA29 (83.3%) (Appendix A). PpIAA26 and PpIAA33 do not contain domain I, and PpIAA26 lacks domain II, suggesting that these proteins may have a non-canonical function compared to other PpIAAs, or participate in different physiological processes than other PpIAAs. In tomato, SlIAA32 lacks domain II, whereas both domains I and II are absent in SlIAA33 [32]. Previous work in *Arabidopsis* demonstrated that domain I of Aux/IAA proteins is an active and portable repression domain containing the ‘LxLxLx’ motif, which interacts with the TOPLESS (TPL) co-repressor [6,7].

Our observations of nuclear localization of fluorescence-labeled peach Aux/IAA proteins support their predicted function as transcriptional regulators. In typical Aux/IAA proteins, there are two NLSs, one is bipartite and the other resembles an SV40-type NLS [12,32]. Our study confirms that representative members of each *PpIAA* sub-clade (PpIAA1, PpIAA5, PpIAA9, PpIAA11, PpIAA13 and PpIAA14) all exhibited nuclear targeting (Figure 3). However, previous research has shown that the natural SlIAA32 protein is present in both the nucleus and the cytoplasm, and thus it is likely that the lack of a bipartite NLS leads to accumulation of SlIAA32 in the cytosol [32], though it remains unclear if targeting to the cytoplasm is also indicative of a potential, unknown extranuclear function.

In peach, several reports have indicated a possible relationship between auxin and fruit development and ripening [33,34,35]. Previous research has demonstrated that the peak of ethylene production occurs during the late stage of peach fruit development, concurrent with an increase in IAA concentration in the mesocarp, thus suggesting a regulatory role for auxin in the control of ethylene biosynthesis [36]. In our study, the IAA concentration was low in the stony hard ‘Jing Yu’ peach cultivar and did not increase at the climacteric stage, however, IAA levels were significantly higher in the melting flesh ‘Okubo’ cultivar than in the ‘Jing Yu’ cultivar (Figure 4). During the late-ripening stage, lower IAA levels in stony hard peaches correlated with lower ethylene production, which may explain why fruit do not soften at maturity in this cultivar. Our findings demonstrate that stony hard peaches are an effective model for investigating the effect of auxin on fruit ripening and softening.

The expression patterns of *PpIAA* genes in various tissues suggest that the encoded proteins may perform both specific and redundant functions. Virtually all 23 *PpIAA* genes were expressed in all assayed tissues, but their expression levels varied considerably. *PpIAA1*, *PpIAA9*, *PpIAA11*, *PpIAA13*, *PpIAA14*, *PpIAA15*, *PpIAA17* and *PpIAA30* may play crucial roles in fruit development and ripening given their higher expression levels in fruit when compared to other tissues, especially *PpIAA33* (Figure 5). The mRNA levels of *PpIAA8*, *PpIAA12*, *PpIAA18*, *PpIAA20*, *PpIAA26* and *PpIAA27* in roots were significantly higher than in other tissues (Figure 5), implying that these genes may play a regulatory role in root development. Previous research has shown that iaa1, iaa12 and iaa28 *Arabidopsis* mutants displayed stronger lateral root growth and apical dominance than wild type [6,37]. In tomato, *SlIAA3* has been shown to participate root growth and tropism [38], and *SlIAA9* is related to fruit set and development [23].

*Aux/IAA* genes are essential factors in auxin signal transduction [39]. Previous research has shown that the increased production of ethylene induced by auxin causes the softening of mesocarp tissue in melting cultivars [2]. Our study therefore compared the expression levels of *PpIAAs* during fruit development between mesocarp sampled from the melting ‘Okubo’ and stony hard ‘Jing Yu’ peach cultivars. The expression levels of 14 genes were higher in ‘Okubo’ than in ‘Jing Yu’ during almost all developmental stages (Figure 6A), which strongly suggested that these genes may be related to auxin signaling during melting peach fruit ripening. However, to determine whether or not, and how those genes contribute to the ripening process of peach fruits requires further investigation. It also remains unclear whether they are redundant or play distinct roles. A previous study indicated that *SlIAA32* is a functional repressor of auxin signaling, and its expression was limited in the breaker stage of tomato fruit development [32].

*Aux/IAA* genes are responsive to auxin induction. The *Aux/IAA* genes in rice, potato, allotetraploid rapeseed, and *Brassica rapa* were up-regulated by exposure to exogenous auxin, but displayed distinct expression patterns from one another [24,40,41,42]. The transcriptional levels of 22 of the 23 *PpIAA* genes were up-regulated by auxin treatment in ‘Jing Yu’ mesocarp, though also to varying degrees (Figure 8). Members of the *Aux/IAA* gene family from *Arabidopsis* and tomato have also been shown to respond to the application of exogenous IAA with distinctly individual expression patterns dependent on time and concentration [11,43]. These differences may be related to tissue-specific auxin perception, differential regulation of free auxin concentrations, or different modes of auxin-dependent transcriptional and post-transcriptional regulation.

Taken together, the *Aux/IAA* gene expression data obtained in this study extends our current knowledge concerning auxin function in climacteric fruits at the ripening stage and indicates several target gene candidates for further exploration of the underlying regulatory mechanisms. Furthermore, a better understanding of the functional divergence of *Aux/IAA* family members in peach, potentially through closer examination of the protein–protein interactions of *Aux/IAAs* with other factors, will provide a valuable resource for molecular selection and engineering for new and desirable fruit phenotypes in peach.

## 4. Materials and Methods

### 4.1. Identification of Aux/IAA Gene Family Members in Peach

All peach proteins in the Genome Database for Rosaceae (GDR) were searched via BLAST + algorithms using *Arabidopsis Aux/IAA* protein sequences as queries [10]. To perform subsequent detailed phylogenetic and structural analyses, all of the obtained sequences were designated as unique genes.

### 4.2. Bioinformatic Analyses of PpIAA Genes

Multiple sequence alignments were generated using ClustalX2 and Espript 3.0 (http://espript.ibcp.fr/ESPript/cgibin/ESPript.cgi) with default parameters. Conserved peptide chains were investigated using Weblogo (http://weblogo.berkeley.edu/logo.cgi) and the MEME tool (http://meme-suite.org/index.html). The isoelectric points (pIs) and molecular weights (MWs) of all translated PpIAA DNA sequences were predicted using the ExPASy Proteomics Server (http://web.expasy.org/compute_pi/). Phylogenetic reconstruction was conducted with the Neighbor-Joining (NJ) method based on the p-distance model of amino acid substitutions in MEGA 5.0 software (5.0; MEGA Inc., Englewood, NJ, USA). Non-parametric bootstrapping was performed with a bootstrap replication value of 1000. Protein subcellular localization was predicted using the predict protein web tool (http://www.predictprotein.org/home).

### 4.3. Plant Materials and Treatment

The fruits of two cultivars, melting textured ‘Okubo’ and stony hard ‘Jing Yu’, were compared in this study. All peach accessions were grown in the experimental orchard of the Beijing University of Agriculture (N40°09′39.45″, E116°30′76.86″, Changping District, Beijing, China). Buds, young leaves, mature leaves, and roots of ‘Okubo’ at the vegetative growth stage, as well as flowers at the full bloom stage, were collected in 2017. Fruit growth stages were defined according to previous research [44]. Growth stages (S1, S2, S3 and S4) were defined by using the first derivative of the growth cumulative curve. During the peach fruit development stages, fruits of ‘Okubo’ and ‘Jing Yu’ were picked at 45, 61, 77, 85, 93, 101 and 109 days (‘Okubo’ fruit matures in 101 days) after full bloom (DAB) in 2017 and quickly delivered to the laboratory for measurement of flesh firmness, as well as for analysis of ethylene and IAA production. Peach tissue was frozen in liquid nitrogen and stored at −80 °C until further analysis. For NAA (1-naphthylacetic acid) and PCIB (p-chlorophenoxyisobutyric acid) treatment of mesocarp discs, ‘Jing Yu’ fruits of uniform size and color in the S4-2 period were used. Tissue cylinders (9 mm in diameter) were excised from 20 fruits with a cork borer, and discs 2–3 mm in thickness were cut from the cylinders using a scalpel. Twenty grams of mesocarp tissue discs were soaked in MS liquid medium [45] (pH 5.5) for 30 min and were then randomly divided into three groups and transferred to 0.5 mM NAA, 20 μM PCIB, or water (control). Samples were collected at 1.5 h, 3 h, 6 h and 12 h. All samples were immediately frozen in liquid nitrogen and stored at −80 °C. This experiment was performed using three replicates.

### 4.4. IAA Concentration, Ethylene Production and Flesh Firmness

The endogenous IAA content was analyzed according to a previously described protocol [46]. Briefly, the mesocarps of peach fruits were ground in liquid nitrogen and 1 g of ground samples was extracted overnight at 4 °C in 80% methanol with sodium diethyldithiocarbamate as an antioxidant. The sample was purified further with a 3-cc Oasis Anion MCX SPE (Waters, MA, USA). The eluted solution was dried under vacuum, dissolved in 50 μL of HPLC initial mobile phase, and filtered through a 0.25 μm filter (4 mm in diameter). The separation was performed using an HPLC-ESI-MS (ThermoFisher, Waltham, MA, USA) and the data were analyzed using Xcalibur 2.1 (ThermoFisher, Waltham, MA, USA). Each analysis was performed in triplicate. To measure the ethylene content of whole fruit, three fruits of each cultivar were kept individually in 940 mL glass jars for 3 h. Gas samples (1 mL) of the effluent air were taken with a syringe and were injected into a gas chromatograph (model 6890N GC; Agilent Technologies, USA). The rate of ethylene production was expressed as μL·g^−1^·h^−1^. Fruit firmness (N) was measured on opposite sides of each fruit using a hand-held penetrometer (Handpi GY-4, 3.5 mm diameter).

### 4.5. Subcellular Localizations of PpIAA-GFP Fusion Proteins

The coding sequence (CDS) of PpIAA1/5/9/11/13/14 were respectively cloned into the pBI121 vector downstream of the CaMV 35S promoter at the multiple cloning site. Gene-specific primers used for PCR are listed in Appendix A. The constructs were transformed into Agrobacterium tumefaciens strain GV3301. Bacterial suspensions were infiltrated into young but fully expanded leaves of 7-week-old tobacco *(Nicotianatabacum* L.) plants using a needleless syringe. After infiltration, plants were grown first in darkness for 12 h and then with 16 h light for 48–72 h per day at room temperature. GFP signal was observed by confocal microscopy (Leica TCS SP5, Leica Microsystems Inc., Heidelberg, Germany).

### 4.6. Quantitative Real-Time PCR (qRT-PCR) Assays

Total RNA was isolated using the EASYspin RNA Rapid Plant Kit (Biomed, Beijing, China). First-strand cDNA was synthesized using the RNase M-MLV kit (Takara, Kusatsu, Japan) according to the manufacturer’s instructions. Three replicates were prepared for each sample. Gene-specific primers for qPCR analysis were designed using Primer Premier 5 software (5.0; Premier Biosoft; Palo Alto, CA, USA) and are listed in Appendix A. Amplification reactions were carried out using Takara SYBR Premix Ex TaqII on an Applied Biosystem StepOne PCR System (48-well). Three independent biological replicates and three technical replicates of each sample were analyzed. The expression levels of *PpIAA* genes from diverse RNA samples were normalized using the translation elongation factor 2 (*PpTEF-2*) gene as an internal control gene. Quantification of mRNA levels was based on the comparative Ct method and was calculated as 2^-ΔΔCt^.

### 4.7. Statistical Analyses

Differences between values were calculated using Student’s *t*-test in Microsoft Excel software (office 2016, Microsoft, Washington, USA), with *p* < 0.05 considered significant and *p* < 0.01 considered very significant. All experiments were repeated at least three times and consisted of three independent biological replicates. The values shown in figures represent the average values of three replicates. Heatmaps were generated in Heml 1.0 (1.0, CUCKOO Workgroup, Wuhan, China).

## Figures and Tables

**Figure 1 ijms-20-04703-f001:**
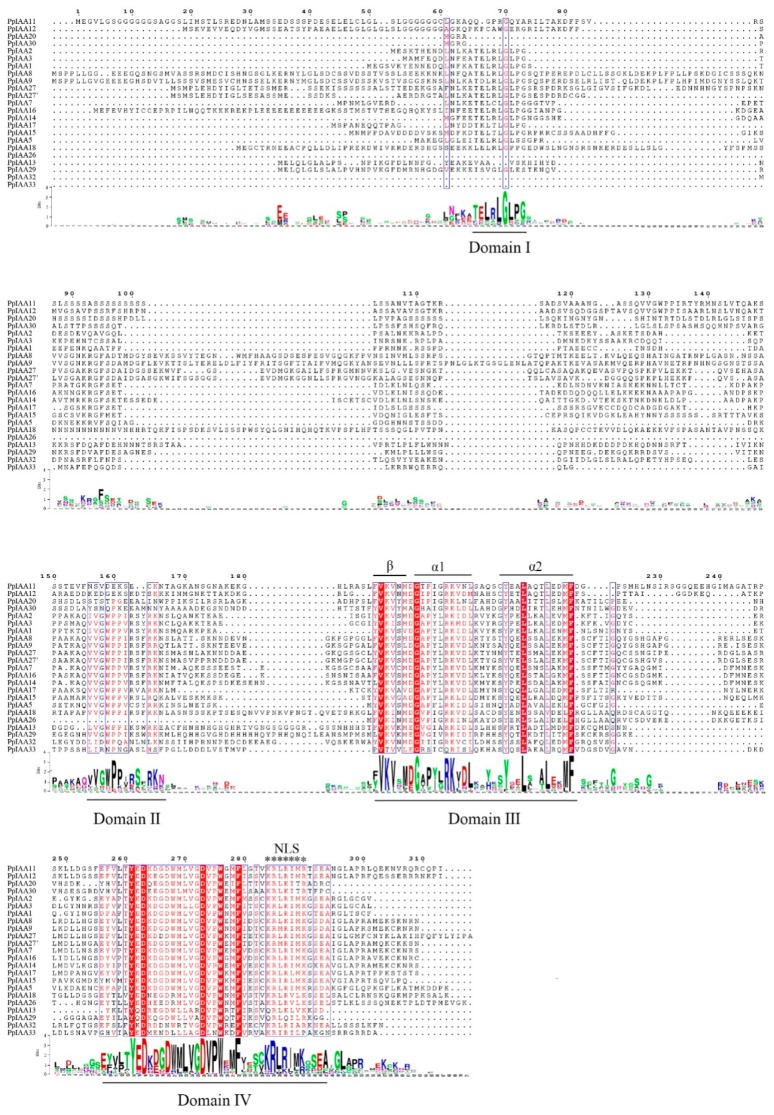
Multiple sequence alignment of PpIAA sequences. The conserved domains (I, II, III, and IV) of the PpIAA family are underlined. Nuclear localization signals (NLSs) are indicated with black asterisks. Bits indicate amino acid conservation at each position. The βαα motif residues in domain III are indicated with “β”, “α1” and “α2”.

**Figure 2 ijms-20-04703-f002:**
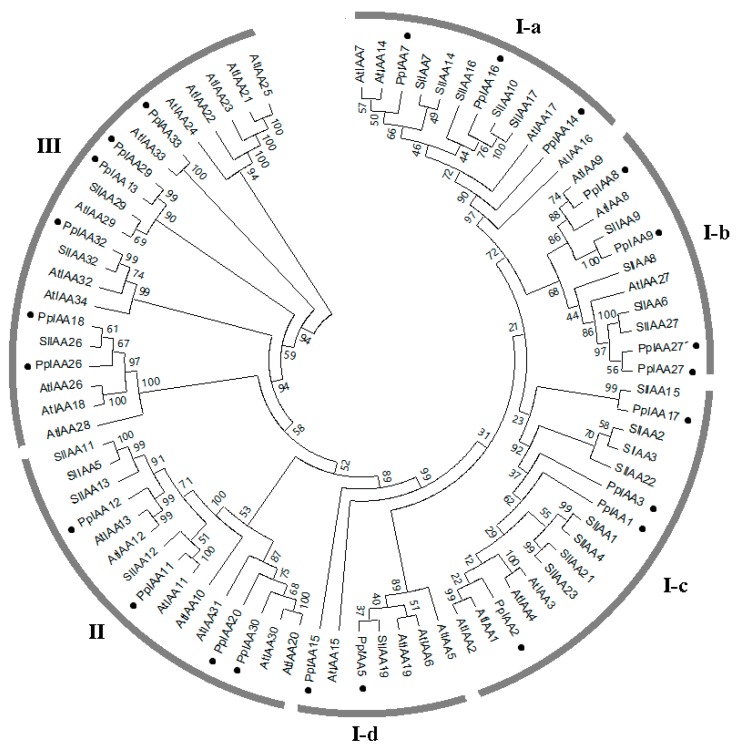
Phylogenetic relationships among auxin/indole-3-acetic acid (Aux/IAA) family members. The Neighbor-Joining (NJ) tree contains 23 PpIAAs, 25 SlIAAs and 34 AtIAAs distributed across 3 major clades, with 4 sub-clades within group I. Bootstrap values were calculated from 1000 replications and are indicated at each node. PpIAAs are indicated with filled black circles.

**Figure 3 ijms-20-04703-f003:**
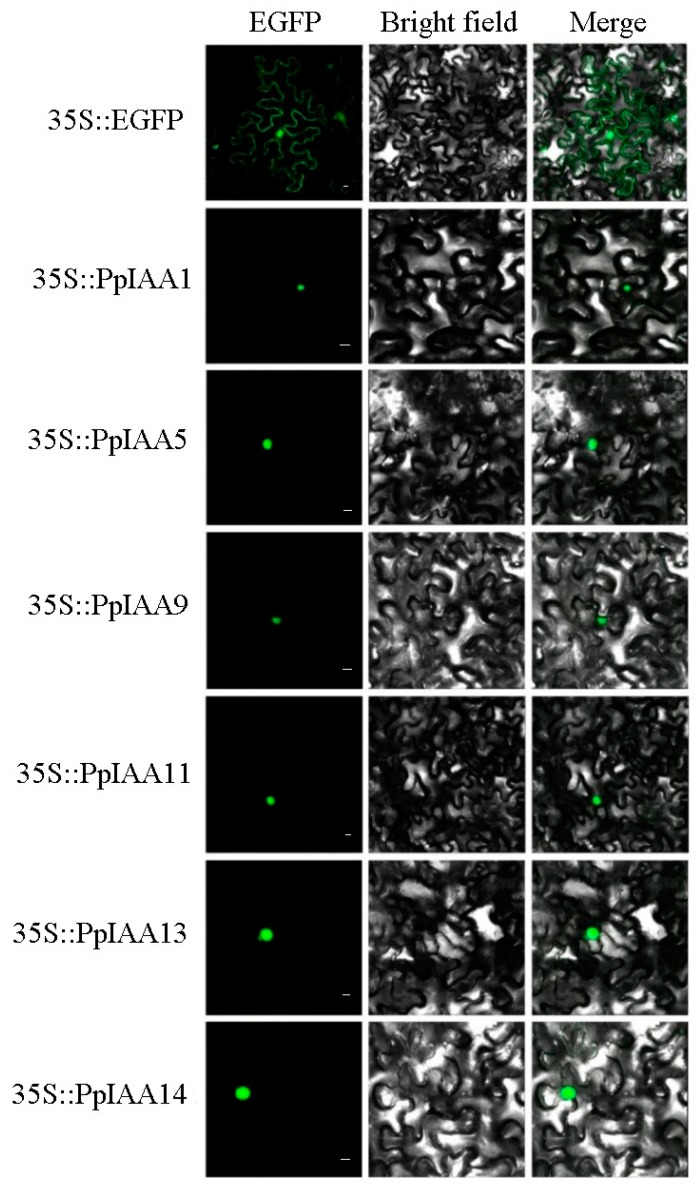
Subcellular localization of selected peach Aux/IAA proteins. PpIAA1-GFP, PpIAA5-GFP, PpIAA9-GFP, PpIAA11-GFP, PpIAA13-GFP and PpIAA14-GFP fusion proteins were transiently expressed in tobacco leaves and their subcellular localization was determined by confocal microscopy. The green fluorescent ball represented the localization of the fusion protein to the nucleus, and the green fluorescent curve represented the localization of the protein to the cytomembrane. The scale bar indicates 15 μm.

**Figure 4 ijms-20-04703-f004:**
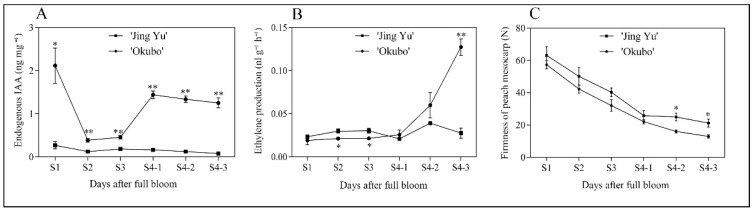
Changes in endogenous IAA concentration (**A**), ethylene production (**B**), and firmness (**C**) in the stony hard ‘Jing Yu’ and the melting ‘Okubo’ peach cultivars. The horizontal axes indicate stages of fruit development: S1, cell division and enlargement; S2, slow growth (pit hardening stage); S3, second period of exponential growth; and S4, the physiological maturity stage, which can be further divided into three stages (S4-1, S4-2, S4-3). Data represent means ± standard error (SE) of three individual experiments. Asterisks indicate statistically significant differences determined using a Student’s *t*-test (* *p* < 0.05, ** *p* < 0.01).

**Figure 5 ijms-20-04703-f005:**
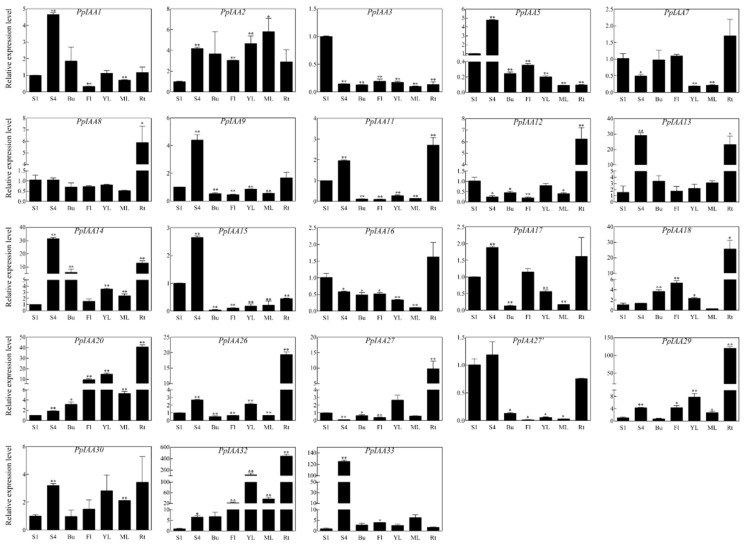
Expression profiles of all 23 *PpIAA* genes in different peach tissues. qRT-PCR analysis of total RNA isolated from fruit stage 1 (S1), fruit stage 4 (S4), bud (Bu), flower (Fl), young leaf (YL), mature leaf (ML), and root (Rt) were used to assess *PpIAA* transcription levels in peach plants. All samples were run in triplicate. Error bars represent the SE of three independent biological replicates. Asterisks indicate statistically significant differences as determined by a Student’s *t*-test (* *p* < 0.05, ** *p* < 0.01). The results were expressed using the fruit stage 1 as a reference for each gene (relative expression level was acted as 1).

**Figure 6 ijms-20-04703-f006:**
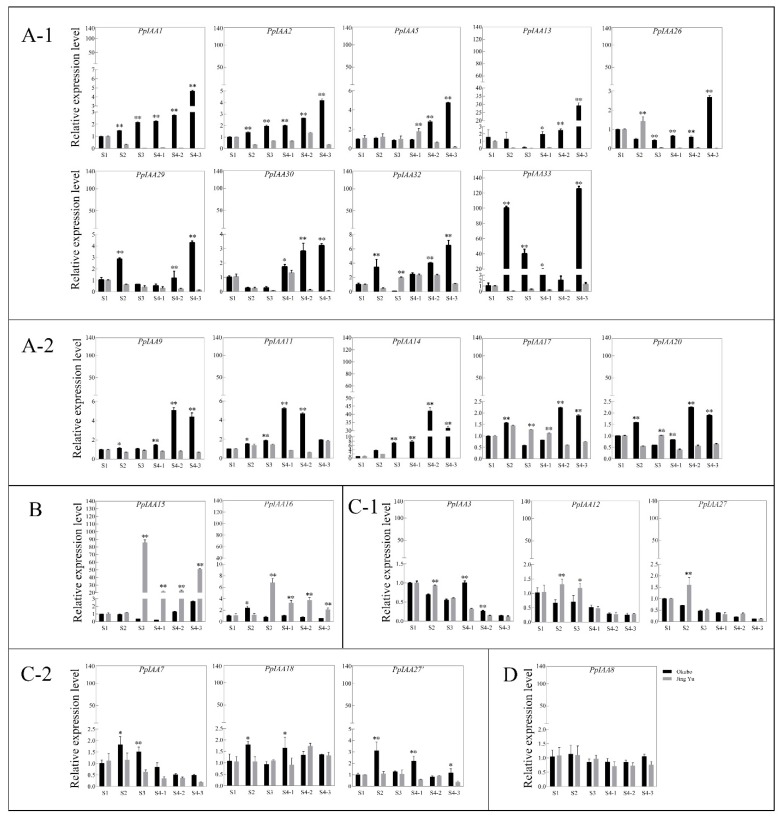
Expression profiles of all 23 *PpIAA* genes during development of ‘Okubo’ and ‘Jing Yu’ peach fruits. RNA was isolated from peach mesocarp at different development stages and was analyzed by qRT-PCR. The horizontal axes indicate stages of fruit development: S1, cell division and enlargement; S2, period of slow growth (pit hardening stage); S3, second period of exponential growth; and S4, the physiological maturity stage, which can be further divided into three stages (S4-1, S4-2, S4-3). Similar expression patterns of genes were divided into four groups: (**A**) the genes were more highly expressed in ‘Okubo’ than in ‘Jing Yu’ in almost all development stages; (**B**) the genes more highly expressed in the stony hard cultivar during ripening stages; (**C**) the genes that displayed lower expression level in all fruit developmental stages and the expression of genes were significantly different in S2 between two cultivar; (**D**) the expression of gene was not significantly different between the two cultivars at any stage. All samples were run in triplicate. Error bars represent the SE of three independent biological replicates. Asterisks indicate statistically significant differences as determined by a Student’s *t*-test (* *p* < 0.05, ** *p* < 0.01).

**Figure 7 ijms-20-04703-f007:**
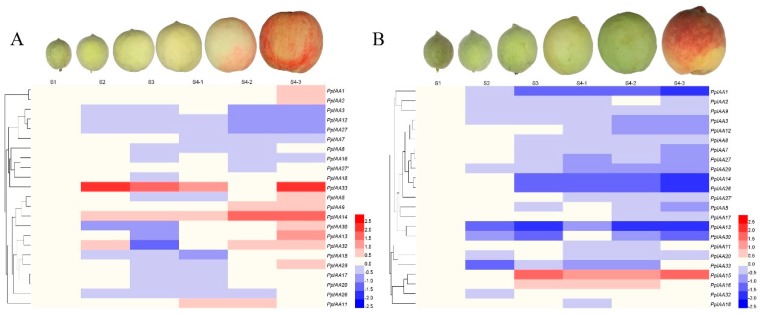
Heatmap of expression profiles of 23 *PpIAA* genes during peach fruit development in (**A**) ‘Okubo’ and (**B**) ‘Jing Yu’ varieties. Heatmaps display hierarchical clustering of average log values of the *PpIAA* gene expression levels during different developmental stages of ‘Okubo’ and ‘Jing Yu’ fruits, respectively. High and low expression are indicated by red and blue, respectively.

**Figure 8 ijms-20-04703-f008:**
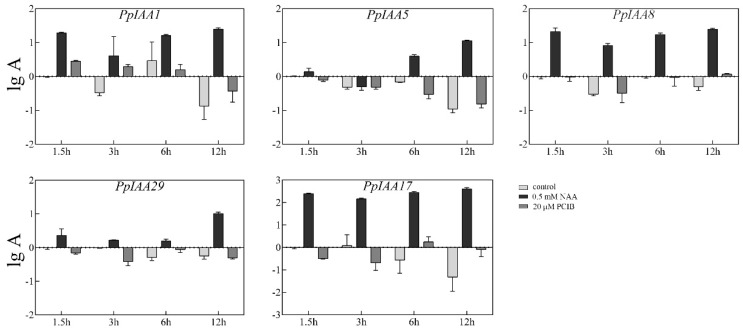
Expression of some of *PpIAA* genes in S4-1 stage ‘Jing Yu’ fruit under NAA (1-naphthylacetic acid) or PCIB (p-chlorophenoxyisobutyric acid) treatment. RNA was isolated from the mesocarp of ‘Jing Yu’ peach fruit following exposure to auxin analogs, and was subsequently analyzed by qRT-PCR. All samples were run in triplicate. Error bars represent the SE of three independent biological replicates. A indicates relative expression level.

**Table 1 ijms-20-04703-t001:** Characteristics of Auxin/indole-3-acetic acid (*Aux/IAA*) family members in the peach genome.

Gene	Genbank ID	Peach Gene ID	Location	ORF Length (bp)	protein Length (Amino Acids)	MW (kDa)	pI
*PpIAA1*	XM_007201785.2	*ppa011843m*	NC_034016.1 (20653629..20654992)	579	193	21.71	5.96
*PpIAA2*	XM_007226200.2	*ppa018535m*	NC_034009.1 (1919205..1920826)	588	196	22.03	6.22
*PpIAA3*	XM_007205878.2	*ppa011755m*	NC_034014.1 (29575406..29576755)	591	197	27.17	7.64
*PpIAA5*	XM_007215941.2	*ppa011935m*	NC_034011.1 (5480855..5484225)	573	191	26.08	7.74
*PpIAA7*	XM_007223748.2	*ppa010698m*	NC_034009.1 (1933829..1936074)	744	248	31.38	8.07
*PpIAA8*	XM_007205297.2	*ppa007194m*	NC_034014.1 (21881754..21886932)	1134	378	45.66	6.75
*PpIAA9*	XM_007225664.2	*ppa006744m*	NC_034009.1 (44167484..44185483)	1098	366	42.56	8.16
*PpIAA11*	XM_007200391.2	*ppa008953m*	NC_034009.1 (44167484..44185483)	939	313	35.43	8.57
*PpIAA12*	XM_007223283.2	*ppa009545m*	NC_034009.1 (3457678..3460446)	1239	413	35.39	8.57
*PpIAA13*	XM_007204663.2	*ppa010871m*	NC_034015.1 (21023201..21025144)	696	232	26.49	9.00
*PpIAA14*	XM_007201338.2	*ppa010342m*	NC_034016.1 (20644191..20646722)	741	247	27.34	6.77
*PpIAA15*	XM_007204658.2	*ppa010303m*	NC_034015.1 (20438534..20440455)	765	255	28.41	8.22
*PpIAA16*	XM_020566397.1	*ppa009254m*	NC_034016.1 (20644191..20646722)	765	255	38.01	5.27
*PpIAA17*	XM_007215909.2	*ppa011570m*	NC_034011.1 (5491090..5492759)	615	205	22.49	7.56
*PpIAA18*	XM_007215529.2	*ppa007663m*	NC_034011.1 (4137359..4141271)	1080	360	44.52	8.86
*PpIAA20*	XM_007225829.2	*ppa011821m*	NC_034009.1 (22238109..22239397)	582	194	21.40	5.48
*PpIAA26*	XM_020556087.1	*ppa013361m*	NC_034009.1 (37689988..37692150)	507	169	19.14	5.99
*PpIAA27*	XM_007215554.2	*ppa007893m*	NC_034011.1 (4585170..4588218)	1029	343	43.21	7.17
*PpIAA27* *′*	XM_020568496.1	*ppa009134m*	NC_034015.1 (19999628..20002859)	918	306	32.41	6.84
*PpIAA29*	XM_007223747.2	*ppa010683m*	NC_034009.1 (40205332..40206887)	720	240	27.03	9.30
*PpIAA30*	XM_020559789.1	*ppa020369m*	NC_034011.1 (141125..143193)	579	193	21.40	5.48
*PpIAA32*	XM_007224433.2	*ppa023002m*	NC_034009.1 (30532207..30533285)	582	194	22.35	4.89
*PpIAA33*	XM_007226683.2	*ppa018956m*	NC_034009.1 (6379308..6380785)	486	162	17.98	9.24

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
