# Peer review of "Genome-Wide Analysis and Identification of the Aux/IAA Gene Family in Peach"

_ijms, 2019, doi:10.3390/ijms20194703_

Round 1

Reviewer 1 Report

General Comments

Genome-Wide Analysis and Identification of the Aux/IAA Gene Family in Peach is about studying  Auxin/indole-3-acetic acid (Aux/IAA) genes role in determining texture and hardness in stone fruit maturation. 23 Aux/IAA genes were identified in this study. Authors concluded that their study provides insights into the evolution of Aux/IAA members in peach and their expression patterns in various tissues and in response to auxin, which can potentially be developed as useful markers for quantitative traits associated with fruit phenotype. Article is interesting but I have following comments before it should be considered for publication in your valuable journal:

Abstract:

It has so many typo mistakes and message is not clear. Its better to modify it in whole.

It should be written by considering following points

Rationale Methodology Results in Quantitative way Conclusion/Recommendations

Introduction:

Problems to be addressed was not highlighted. Similarly, so many typo mistakes and words are joined with each other. See comments on main file

Objective of study not mentioned.

Results

Again, so may typo mistakes.

Figures quality should be improved.

See Comment in main file.

Discussion

Written in good way but again so many typos here. See main file with comments

Materials and Methods

Add Ref for BLAST+

Multiple sequence alignments were generated using ClustalX2 and Espript 3.0 http://espript.ibcp.fr/ESPript/cgibin/ESPript.cgi) with default parameters. Why default parameters were used and what are these?

Add coordinates for Beijing University of Agriculture site.

Reviewer 2 Report

In this manuscript, the authors attempted to characterize a family of Aux/IAA genes that were identified in the genome of peach. Specific expression of PpIAAs in time- and tissue-dependent manner and sensitivity of PpIAA1/8/17 to NAA treatment suggest that these genes are involved in auxin signaling in peach. The authors also speculate that some of these genes may be related to peach fruit ripening, but do not show any mechanistic evidence to support their claim. The major issue of this paper is that it does not bring any real comparison with Aux/IAA genes from classical model organisms – the phylogenetic analysis is only briefly mentioned. Further, no functional analysis of PpIAAs genes had been performed to convincingly link the function of those genes with observed developmental processes or endogenous IAA profile changes. Presented data thus do not seem to be much of a novel finding and in my opinion do not meet criteria for publication in IJMS. There are also several smaller issues:

- localization of PpIAAs tagged with GFP is presented with low-quality images: nuclei not visible in BF and no control staining (e.g. DAPI staining), scale bar missing in all panels. Figure 3 legend incomplete (what kind of GFP fusions?...). No prove of PpIAA-GFP constructs correct expression – e.g. WB analysis to confirm the functionality of those new GFP constructs.

- in Figure 5 relative expression is shown. However, it not clear how the expression values are related to one another (expression level 1.0?); no statistical analysis mentioned

- Materials and Methods part is not divided into sections  

- a large number of typos

Reviewer 3 Report

the Manuscript is interesting

the only problem the English and style mostly spacing throughout the whole manuscript

for Example 

Line 21 please change genomeusing to genome using

Line 24 exhibitdifferential to exhibit differential

this type of mistake is everywhere in the manuscript so please double check the manuscript for this kind of mistakes

Round 2

Reviewer 1 Report

Authors have incorporated all suggestions so it should be considered for Publication.

Author Response

Thank you for your hard work. 

Reviewer 2 Report

The response to the reviewer and the revised version of the manuscript show that the authors tried to dealt with all major suggestions and considerably improved the text. I only noticed minor typographical errors (name of genes should be in italics, check e.g. the Abstract). I have only one question: this again concerns Fig 5. The information about statistical analysis is still missing in figure legend. More importantly, there are no asterisks in the figure that signify statistical change (as in Fig. 6). Yet some changes such as those of PpIAA33 seem to be very significant. This should be clarified.
